# Blockchain Token-Based Wild-Simulated Ginseng Quality Management Method

**DOI:** 10.3390/s22145153

**Published:** 2022-07-09

**Authors:** Youngjun Sung, Sunghyun Yu, Yoojae Won

**Affiliations:** Cyber Security Laboratory, Department of Computer Engineering, Chungnam National University, Daejeon 34134, Korea; tjd428@o.cnu.ac.kr (Y.S.); yoursaint@cnu.ac.kr (S.Y.)

**Keywords:** blockchain, smart contract, blockchain token, quality management

## Abstract

Countries require measures to prevent food fraud, such as forgery of certificates or content change during production, which can occur throughout the supply chain, even if they have a certification system for quality food management. Therefore, there are recent cases of the introduction of blockchain tokens for quality and supply chain management; however, there are difficulties in introducing tokens in food fields, such as forest and agricultural products. To introduce tokens in the food sector, we selected wild-simulated ginseng, subject to quality management in Korea, analyzed the quality management process of wild-simulated ginseng, and selected the target for blockchain token introduction. We then identified potential token-related issues from consumers and suggested possible solutions.

## 1. Introduction

Although the interest in organic products is increasing worldwide, it is difficult for consumers to directly verify foods in the market as organic; therefore, several countries have enacted specific laws or standards to manage and certify organic foods [1]. In Korea, a quality management system has been legally established for a food called wild-simulated ginseng, which is, step by step, managed from production to distribution by the government. The government provides certificates for each step passed. Producers must submit documents from the previous step to pass to the next. However, this process is executed offline using paper documents for a very long time, making it difficult to ensure the integrity of information and incurring high document management costs. In addition, fraud may occur, such as forging the offline certificates or changing the contents. [2,3].

Now, IoT and big data are being used to collect and manage supply chain information regarding agricultural products [4,5]; however, most of these tools cannot sufficiently ensure high integrity of this information [6]. Blockchain technology provides a better chance to achieve the desired high integrity; the technology has proved helpful to various industries, such as finance [7,8,9]. Blockchains comprising hash chains provide transparent supply chain management processes and improve data integrity. Nonfungible tokens (NFTs) are also becoming popular in authentication and process management [10,11,12,13]. NFTs facilitate tracking the physical or non-physical possessions of unique, non-exchangeable tokens (such as digital art or luxury apparel). However, because food is an item that disappears when ingested, information asymmetry may occur because the objects that the tokens track are destroyed.

Nevertheless, utilizing blockchain and NFTs technology, we designed a reliable wild-simulated ginseng management system. The system tracks wild-simulated ginseng from planting to the market. Legal documents—tokens are generated at each process step and tracked thoroughly. If a token is managed instead of the existing offline documents, the existing offline document management system can be replaced. Indeed, a token verifying that some wild-simulated ginseng has passed all the quality tests can be passed on to the end consumer. A gateway contract ensures that only valid tokens are circulated in the market. Distribution is blocked if all token transfers do not pass the logic of the gateway contract.

## 2. Related Work and Background

This chapter provides background information on related research and proposals for blockchain tokens.

### 2.1. Blockchain Token Based Supply Chain Processes

The authors in [10,11] proposed a blockchain-based supply chain tracking system using contracts in Ethereum virtual machines. They defined smart contracts as recipes by manufacturers, referring to the product composition. Each recipe component is managed using an NFT and matches the product.

The author in [12] proposed a multi-agent system (MAS) architecture on a blockchain basis for a transparent agri-food supply chain. They manipulated the blockchain through smart contracts by placing various agents for each layer and tokenizing agricultural products sold on the platform per the ERC-721 standard for irreplaceable tokens.

A supply chain tracking system using smart contracts on the blockchain to prevent malicious producer fraud requires third-party certification (TPC) through an independent organization that can evaluate and verify compliance with producer standards or legal requirements. The author in [13] implemented TPC broadcast via an ERC-1155 standard-based NFT, enabling reliable supply chain transparency.

Table 1 compares the existing studies. All existing studies have secured traceability and transparency using blockchain tokens in the supply chain. In [10,11], tokens were designed not to be distributed to end consumers, but this can be a problem if you need to provide a certificate to the consumer at the time of sale. The Authors in [12] changed the status value of tokens outside the supply chain but did not incinerate them separately; this will likely continue the token transfer, even if the status value changes. The Authors in [13] extended the characteristics of ERC-721 used in other studies by using ERC-1155. In their study, tokens are passed to consumers, but the authors do not address residual tokens left after product consumption. Summarily, existing studies lack measures to prevent invalid tokens from being distributed. Therefore, an attempt is needed to prevent the distribution of invalid tokens in advance.

### 2.2. Blockchain

A blockchain creates blocks at regular intervals to manage the data. Each new block includes the previous block’s hash value in its header. These blocks are continuously generated over a certain period, and blocks, except for the initial block, form a hash chain with the previous block’s hash value, making it difficult to forge and tamper with blocks [14].

Although there are differences in the header field for each blockchain, such as Bitcoin and Ethereum, Figure 1 shows the general blockchain header structure. The n-th block includes the hash value of the previous block (N − 1) header in the previous hash field, and the n-th block header hash value is included and recorded in the block (N + 1) added thereafter. The block-creation time is recorded in the timestamp field, the Merkle tree [15] is constructed using the transactions included in the block body in the root hash field, and the generated Merkle root value is recorded. The n-th block root hash field in Figure 1 was recorded, assuming that four transactions were included in the body.

These generated blocks are distributed by peer-to-peer (P2P) network participants and are selected to be added using consensus algorithms, such as proof of work (PoW), proof of stake (PoS), and practical Byzantine fault tolerance (PBFT) [16].

### 2.3. Smart Contract and Account

A smart contract is a concept proposed by Nick Sabo that automatically executes a contract written in a program code when certain conditions are satisfied [17]. A smart contract was implemented in Ethereum, developed by Vitalik Buterin, and has been applied in various fields [18].

In Ethereum, there are two accounts: an externally owned account (EOA) and a contract account (CA) [19]. An EOA is an account used by general users, and private and public keys exist to enable the creation and execution of transactions. Figure 2 shows the process of generating the EOA and creating a 256-bit random value as a private key. From the secp256k1 curve, the elliptic curve digital signature algorithm (ECDSA) [20] generates a public key, which is a replica of the private key. The generated public key is used to obtain a hash value using the Keccak-256 algorithm [21], and the lower 160 bits of the hash value are used as the EOA. A CA is an account created when the smart contract code is compiled and deployed on the blockchain. Functions can be called and interact within the contract through the account, and the CA cannot create a transaction alone. Therefore, no private and public key exists.

### 2.4. Transaction Type

Ethereum transactions can be divided into external and internal transactions [22]. An internal transaction cannot be created by itself and the internal transaction logic implemented inside the smart contract must be executed by an external transaction. For external transactions, the EOA can sign a transaction and send it to the EOA or CA to call a function. In Figure 3, CA 1, called the EOA, calls a function to CA 2 by using internal transactions. The EOA can transmit transactions via the CA, as shown in Figure 4.

### 2.5. Blockchain Token Standard

Table 2 categorizes token standards using a blockchain platform. Ethereum and Klaytn standardized the blockchain token standard to specify functions and provide guidelines. Klaytn is a blockchain created by forking the Byzantium version of Ethereum and has a similar token standard because it forks Ethereum into a blockchain with a block-creation time of 1 s by transforming the PBFT-based algorithm [23].

Tokens can be broadly classified into fungible tokens [24,25], NFT [26,27], and multi-tokens [28,29]. Figure 5 shows the characteristics of the tokens created when a contract is implemented based on the token standard. For a fungible token, one token is always the same as another and can be used in fiat currency and voting rights. NFT is a token that represents the ownership of unique items. In the figure, it can be observed that different tokens have been issued to the NFT contract as A, B, C and D. A multi-token combines the characteristics of the previous two tokens, and multiple tokens can be represented in a single contract. In the figure, tokens A, B, C and D are issued in one multi-token contract, and among them, a number of A and C tokens, such as fungible tokens, are issued, and only one B and D tokens, such as NFT, are issued; therefore, they have unique characteristics.

### 2.6. Wild-Simulated Ginseng Quality Management Process

Two government agencies cooperate in implementing the wild-simulated ginseng quality management system in Korea. Legal documents are issued upon passing every one of the four steps in the process. Before proceeding to the next step, documents of the previous step must be produced [30]. However, this process is currently executed offline using paper documents, making it difficult to ensure the integrity of information and incurring high document management costs. In addition, fraud may occur, such as forging the offline certificates or changing the contents. Therefore, utilizing blockchain and NFTs technology, we designed a reliable automated wild-simulated ginseng management system [31]. Several legal forms exist at each step of the wild-simulated ginseng quality management process. By the last step, the producer would have been issued one quality inspection certificate and 100 pass certificates. When the wild-simulated ginseng is sold, consumers can verify whether it has passed all the quality inspection steps [30]. Figure 6 shows the existing wild-simulated ginseng quality management system consisting of four steps.

## 3. Quality Management Using Blockchain Token

In this study, a blockchain was used to improve the integrity of wild-simulated ginseng quality management. Token standards were selected for the introduction of tokens, and gateway contracts were deployed to prevent food fraud caused by the reuse of the issued tokens or incorrect distribution tokens.

### 3.1. Token Standard Selection

Tokenization is selecting a token standard based on the characteristics of the target. Here, the token standard was selected using the checklist presented in Figure 7, and the standard was selected based on step 4. In the previous step, all documents satisfy the NFT characteristics, but only one quality inspection certificate with unique characteristics is issued in step 4. However, pass certificates are issued in multiple copies and have no unique characteristics. Therefore, a multi-token was selected as the token standard for quality control. The possible token standards can be verified as a metric using the checklist shown in Table 3.

### 3.2. Multi-Token-Based Quality Management

A token implementing a smart contract using a multi-token standard provides functions, such as token transfer and ownership inquiry, based on the interface provided by the standard. Table 4 lists the functions in the multi-token standard ERC-1155 interface provided by Ethereum. Batch inquiries and consistent transfer functions are included because various tokens are managed in a single contract.

A multi-token can be utilized, as shown in Figure 8. Producers receive pass-certificate tokens as they pass the steps of the system. When selling wild-simulated ginseng, certificate tokens can be sent to consumers and used as quality assurance certificates. The consumer can inquire into other tokens to check the quality inspection details. Additional tokens can also be issued with agency approval.

### 3.3. Gateway Contract

Because these tokens are issued for physical foods, there are difficulties in their management compared to digital assets. Malicious users can exploit these problems by validating their tokens using gateway contracts. For this reason, tokens must be transmitted through the interface provided by the multi-token contract. The process is configured to pass through the gateway contract before calling the multi-token contract. If the token is invalid, it is incinerated and cannot be circulated to prevent criminal activity in advance. If the token is valid, the transaction is transmitted to a multi-token contract using internal transactions. Figure 9 shows a case where a malicious user attempts to use an invalid token from producer C. After receiving validation from the gateway contract before using the multi-token contract interface, if it is not valid, it is transmitted to the null address used as the token incineration address to prevent circulation. The corresponding null address was 2.1.2. In the account described in this section, all 160 bits are set to zero; thus, 0 × 0000 … 000 accounts.

### 3.4. Wild-Simulated Ginseng Quality Management Using Blockchain Token

Figure 10 shows the case of combining the aforementioned ERC-1155 and gateway contract with the existing wild-simulated ginseng quality control. In four steps, the results are issued and utilized as blockchain tokens, and tokens distributed together as quality certification mean that wild-simulated ginseng is distributed to consumers after passing all the steps.

Step-by-step details are as follows:Step 1: The producer requests a conformity check from Agency 1. Agency 1 deploys an ERC-1155 contract for the producer and uses the constructor function to minimize the first result token (other than token transfer, it does not require a gateway contract).Step 2: Submit a previously issued token to agency 2, which performs the verification. After receiving the token, the agency issues a second result token if it is suitable for proceeding to the next step. At this time, the token transmission must pass through the gateway contract.Step 3: This is similar to step 2 and is executed by agency 1. If appropriate, the third token result is minted.Step 4: Upon passing as the final step, agency 1 mints the last result token and certificate token to be used for sale.Sale: Consumers can inquire about token ownership when purchasing wild-simulated ginseng. The producer must provide a certificate token through the gateway contract when selling wild-simulated ginseng to consumers. The distribution is blocked in advance if it is an invalid certificate token.

## 4. Gateway Contract Usage Scenario

In this section, the malicious actions that can be prevented using the gateway contract are explained based on possible scenarios.

### 4.1. Scenarios with Timeout Logic in Gateway Contract

In the first scenario, the producer was malicious. Quality certification has an expiration date, after which it must be re-inspected. However, the creator may sell expired products maliciously. To prevent this, a timeout logic could be created in the gateway contract such that tokens that have passed their expiration date are not circulated but sent to the null address and burned. Figure 11 shows that when a timeout occurs, the token burns. Otherwise, the token is sent to the consumer using a multi-token contract.

### 4.2. Scenarios with Threshold Logic in Gateway Contract

The second scenario occurs when a malicious user resells a product different from the target when the token is being tracked. To solve this problem, the characteristics of the tokenization target are identified, a threshold value is set to change the token owner, and when the value exceeds, it is transmitted to the management agency to prevent circulation. In the case of wild-simulated ginseng, direct sales account for more than 90%; therefore, the threshold is set to 1 or 2 to make resale impossible after purchase. Subsequently, a management agency may decide whether to incinerate or allow distribution. Figure 12 shows that the token is sent to the management agency when the threshold is exceeded. Otherwise, the token is sent to the consumer using a multi-token contract.

### 4.3. Scenarios in Bypassing the Gateway Contract

In the third scenario, a malicious user directly submits a transaction to a multi-token contract without going through the gateway contract. The blockchain can verify the address of the user who sends the transaction. In terms of solidity, the sender can be identified using a global variable called msg.sender. If it is not an internal transaction sent by the gateway contract, the msg.sender is recorded as an account other than the gateway contract, and by adding logic to the multi-token contract to identify whether the transaction sender is a gateway contract, the gateway contract is prevented. Figure 13 shows a transaction failure scenario when a malicious user attempts to bypass the gateway contract.

### 4.4. Gateway Contract Logic

Figure 14 shows the token transfer code for the gateway contract. The request is divided into four branch sentences, excluding the conditional sentence, and the contents of each branch point are as follows:Timeout logic: If the token has exceeded the valid time, the token is sent to a null address and incinerated by comparing the issuance time with the current blockchain.Producer logic: On the certificate token, the producer, the first owner, records information on the target sold to the consumer when selling and increases the number of transactions by one.Threshold logic: When a consumer purchases wild-simulated ginseng, receives a token and then attempts to transmit the token, the token is transferred to the management agency account if the number of transactions exceeds the threshold.C2C logic: Transmission of consumer-to-consumer (C2C) records information to the token owner increases the number of transactions when the transmission is possible through previous logic.

## 5. Experiment

The experiment was conducted assuming that step 3 has been passed, and in step 4, we examine the token issuance to the final distribution step. In the distribution step, we implemented and tested three scenarios that could be prevented using the gateway contract.

### 5.1. Environment

The code implemented in Openzeppelin was used to issue pass and quality inspection certificates as ERC-1155. Part of safeTransferFrom in the code is modified to be subordinate to the gateway contract. Because the gateway contract must manage the CAs of multiple producers, in Table 3, in addition to the parameters of the safeTransferFrom function, tokenContract is added such that the CA can be identified to call. Standard functions other than transfers were not included in the experiment.

The blockchain network was constructed using Rinkeby, an Ethereum test network. When using a test network, using a blockchain explorer called Etherscan is advantageous.

Table 5 lists the four EOAs and two CAs used for token quality management and scenario validation. Here, ‘Admin’ is the EOA of a quality management agency. ‘Producer’ is the EOA of the producer. ‘Consumer’ is the EOA of the consumer purchasing wild-simulated ginseng from the producer. ‘Attacker’ is the EOA of a malicious consumer distributing invalid tokens. ‘Gateway contract’ is a CA that must be passed for token transfer. ‘Multi-token contract’ is ERC-1155 CA for managing wild-simulated ginseng products that have passed quality management.

### 5.2. Results

After issuing tokens that can be generated in step 4, we enter the distribution step, and the gateway contract with ERC-1155 is tested. We tested three scenarios, typical token transfers likely problematic, and then identified them through Etherscan.

#### 5.2.1. Minting and Tracking Tokens

Before the malicious scenario test, the 100 certificate tokens generated in Step 4 were issued. Figure 15 shows that tokens are issued, and information about the owner is recorded on the blockchain. As such, blockchain tokens record and provide transparent management records to participants by recording and providing details of the token transfer in the blockchain when the owner is changed. Transactions that change the status of the blockchain require gas costs, but the act of inquiring about the information recorded in the blockchain is free of charge, so many participants can check the history. Therefore, even before receiving the token, the previous history can be referred to through the information recorded in the blockchain, and various user interfaces can be configured using a library such as web3.js that provides an API to interact with the blockchain node.

#### 5.2.2. Timeout Scenario Result

This result occurs when tokens are transferred while selling wild-simulated ginseng, which exceeds the validity period of the quality inspection.

Table 6 lists the parameter data that the product sends to the gateway contract to send the pass token (ID:1) to the consumer. The parameters include multi-token contract CA, product EOA, and consumer EOA.

In Figure 16, the product sends the token, but the token expires, and the gateway contract does not send the token to the consumer but sends it to the null address. This is because the consumer does not receive tokens; therefore, the transaction of wild-simulated ginseng ceases. The malicious producers examined in this section can prevent the sale of wild-simulated ginseng that has expired.

#### 5.2.3. Scenario Results Exceeding Thresholds

This result occurs when tokens are transferred while selling wild-simulated ginseng, which exceeds the threshold for the number of transactions specified by the management agency.

Table 7 lists the parameter data the attacker sends to the gateway contract to send the pass token (ID:5) to the consumer. The parameters include multi-token contract CA, attacker EOA, and consumer EOA.

In Figure 17, although the attacker sends the token to the consumer, the token is sent to the Admin, not the consumer, because it exceeds the threshold set in the gateway contract. Timeout tokens are immediately incinerated upon circulation; however, tokens exceeding the threshold are sent to the management agency to decide whether to allow them back into circulation. This can block malicious actions, such as reselling wild-simulated ginseng, using the tokens provided at the time of purchase by consumers. These tokens are, however, not entirely burned, considering other consumers who want to transfer ownership without malicious intent.

#### 5.2.4. Bypass Scenario Result

This result occurs when the token transfer function is called directly to the multi-token contract without going through the gateway contract.

Table 8 lists the parameter data the attacker may send to the multi-token contract to send the pass token (ID:5) to the consumer. Because it does not go through the gateway contract, multi-token contract CA is not included, and the attacker EOA and consumer EOA are included in the parameters.

In Figure 18, it can be observed that when an attacker executes a token-transfer transaction directly to the multi-token contract, it fails. This means that even if a malicious user attempts to bypass the gateway contract to avoid token validation, their token transfer cannot be successful.

#### 5.2.5. Gas Results

To compare the gas required by the blockchain model in this study, three types of ERC-1155 tokens were tested. First, ERC-1155A is the plan proposed in this study with a gateway contract and dependence. Although contract B is not dependent on the gateway contract, the four branch statements used by Gateway were applied in a similar manner. Contract C is a contract that used ERC-1155 without modification to the existing Openzeppelin. Table 9 shows the gas fees that appear when three functions are executed for the corresponding contracts.

## 6. Discussion

Wild-simulated ginseng quality management using blockchain tokens provides transparent information, improves data integrity, and prevents damage caused by invalid products on the market. Existing studies that have investigated the same problem do not provide tokens to the end consumer as quality inspection certificates; even if they do, they do not deal with risk factors such as residual tokens. In [10,11], tokens were managed by placing smart controls for each process, but the authors did not provide tokens to end consumers as they judged them to be outside the supply chain. This study provides end consumers with tokens through guarantees or certificates. The method proposed by the author in [13] uses the ERC-1155 token, as in our study. However, the study allows third parties in the supply chain, thereby increasing the possibility of circulation of invalid tokens in the market. We devised a system that rejects invalid tokens in advance, thereby increasing the integrity of the process.

Our system includes a contract in the transmission step; thus, compared to normal contracts, it can incur higher gas costs. However, by distributing the verification logic, we can see an inevitable reduction in deployment costs for the ERC-1155 contract. This is advantageous if smart contracts are provided to multiple users. However, when configured as 1:N, the centralization obtained will likely provide a single point of failure (SPOF). Therefore, it is necessary to accommodate contracts by distributing them in an N:M instead of a 1:N configuration.

First, a target must be selected to secure integrity and transparency through the blockchain. This may be a legal document generated during the production, distribution, or information collected using IoT devices. Second, the information recorded in the blockchain is analyzed and tokenized to be managed through a token. Tokens can track changes in ownership to identify the current status of each step; although not covered in this study, large amounts of data, such as images and audio, can be stored and managed using a distributed file system, such as IPFS. However, because tokenization used to manage physical objects can destroy or change the tracking of tokens leading to incorrect tokens, additional measures should be taken to prevent invalid tokens from being distributed in the market.

## 7. Conclusions

The current wild-simulated ginseng quality management system in Korea is offline and uses paper documents, making it difficult to ensure the integrity of information and incurring high document management costs. In addition, fraud may occur, such as forging the offline certificates or changing the contents. A sound wild-simulated ginseng quality management system must ensure high transparency and integrity to gain the trust of consumers or participants. Blockchain technology has been applied to various processes to achieve high certification reliability. Therefore, utilizing blockchain and tokens technology, we designed a reliable automated wild-simulated ginseng management system in Korea. When introducing tokens, the target of tokenization was selected, and an appropriate token was selected by comparing and analyzing the token standard. Because food disappears when consumed; problems may occur when malicious consumers try to reintroduce tokens of consumed wild-simulated ginseng into the supply chain. Here we propose measures to prevent the circulation of these tokens in the market.

In the future, further research is needed to introduce the model to foods other than wild-simulated ginseng. Moreover, a similar framework can be applied to other industries by improving the research focused on the wild-simulated ginseng management blockchain framework.

## Figures and Tables

**Figure 1 sensors-22-05153-f001:**
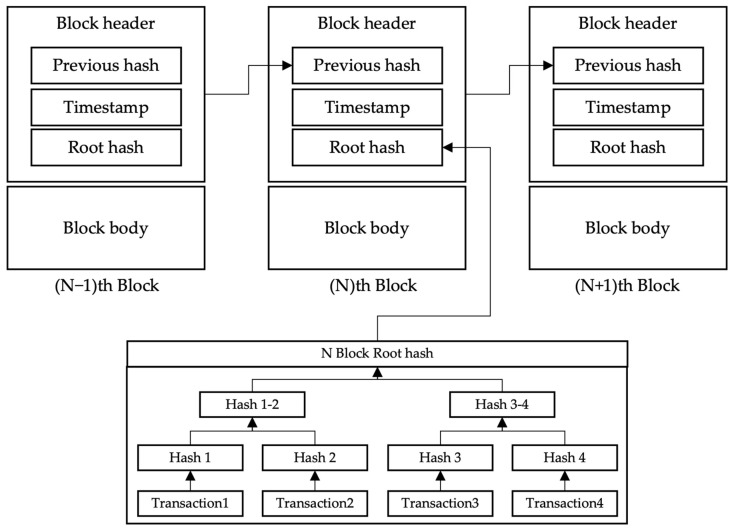
Blockchain Header Structure.

**Figure 2 sensors-22-05153-f002:**
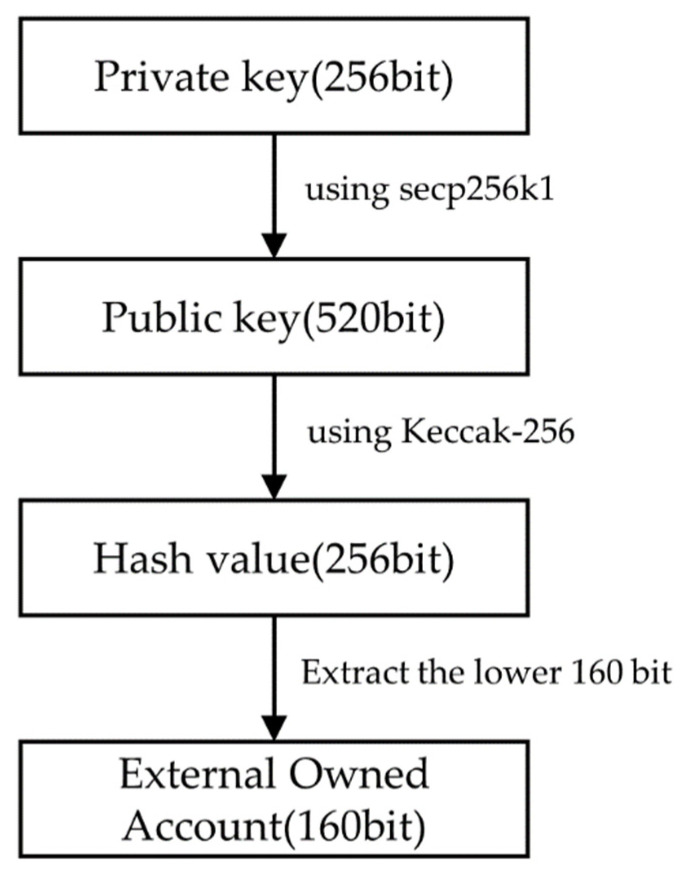
EOA creation process.

**Figure 3 sensors-22-05153-f003:**
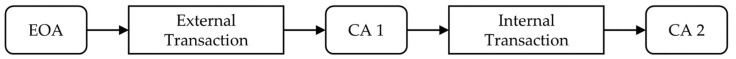
Calling another CA using an internal transaction.

**Figure 4 sensors-22-05153-f004:**
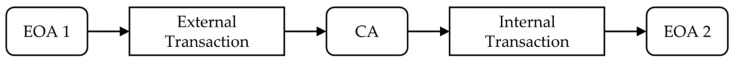
Sending a transaction to another EOA using an internal transaction.

**Figure 5 sensors-22-05153-f005:**
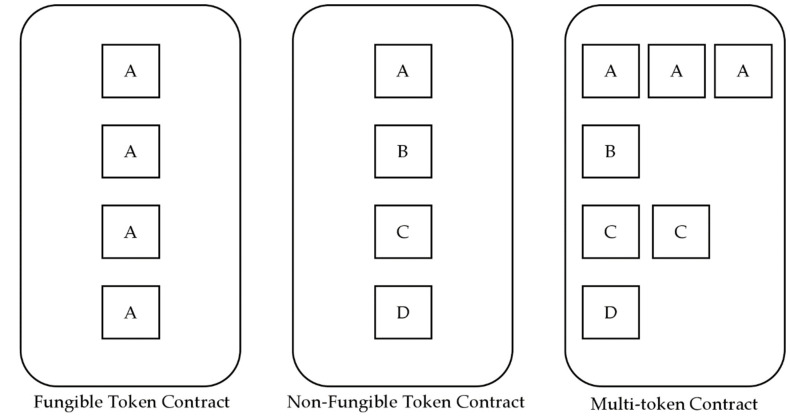
Characteristics of blockchain tokens by standard.

**Figure 6 sensors-22-05153-f006:**
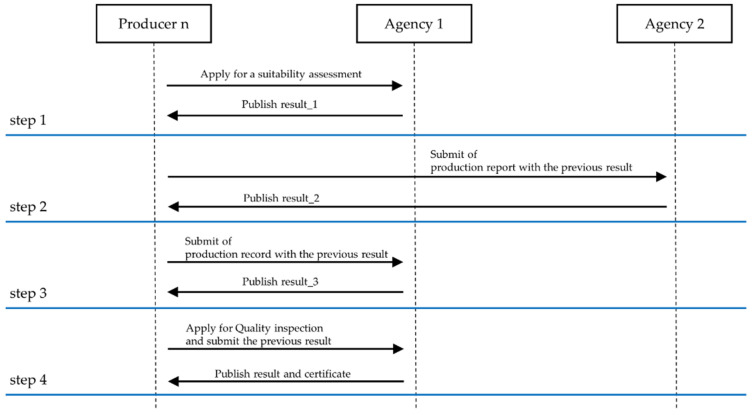
Wild-simulated ginseng quality management process sequence diagram.

**Figure 7 sensors-22-05153-f007:**
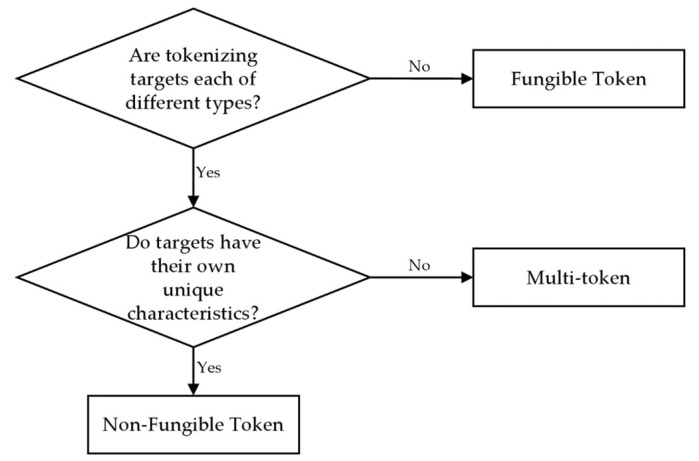
Checklist when selecting token standards.

**Figure 8 sensors-22-05153-f008:**
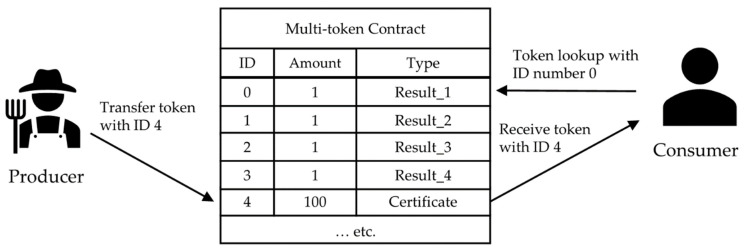
Quality management using multi-token.

**Figure 9 sensors-22-05153-f009:**
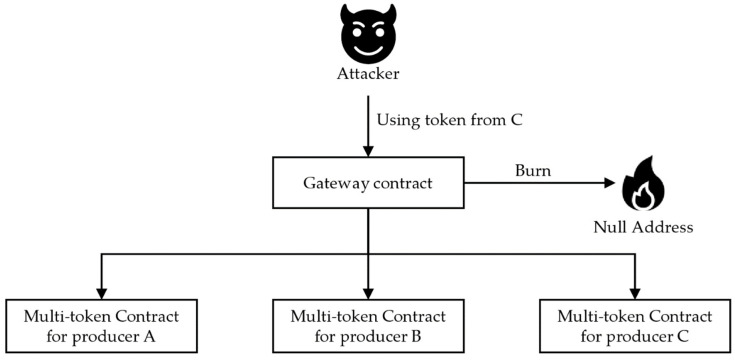
Usage of a gateway contract to burn invalid tokens.

**Figure 10 sensors-22-05153-f010:**
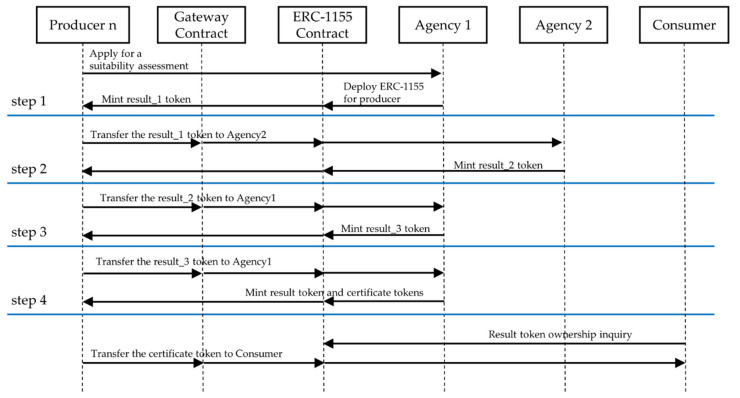
Blockchain-based wild-simulated ginseng quality management process sequence diagram.

**Figure 11 sensors-22-05153-f011:**
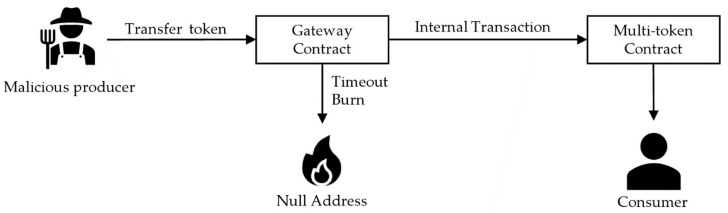
Timeout logic included in gateway contracts.

**Figure 12 sensors-22-05153-f012:**
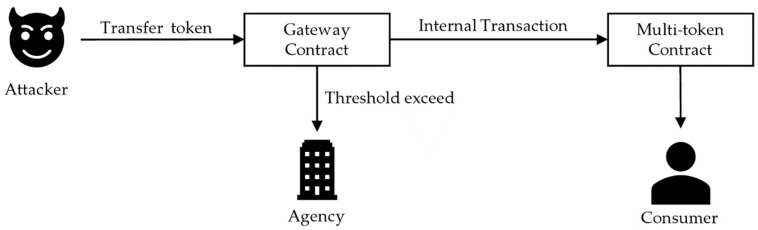
Threshold logic included in gateway contracts.

**Figure 13 sensors-22-05153-f013:**
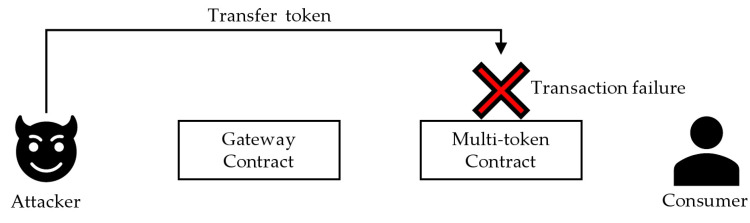
Transactions that do bypass the gateway contract fail.

**Figure 14 sensors-22-05153-f014:**
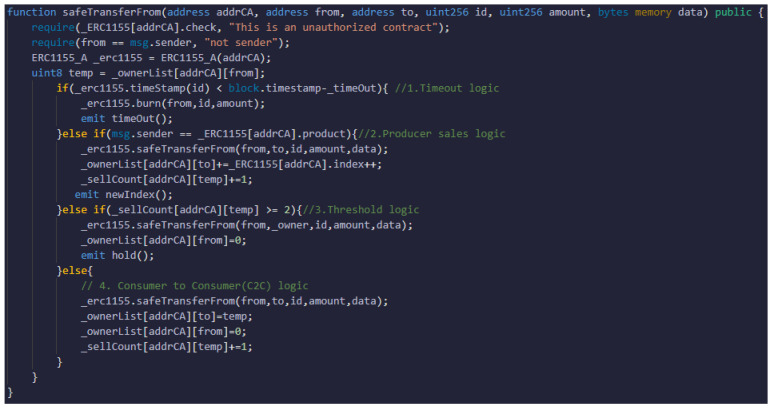
Token transfer code in Gateway contract.

**Figure 15 sensors-22-05153-f015:**
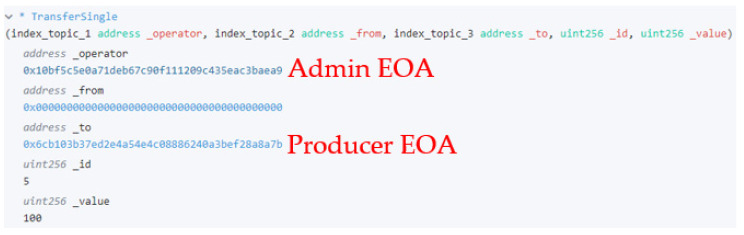
Blockchain token transfer log.

**Figure 16 sensors-22-05153-f016:**
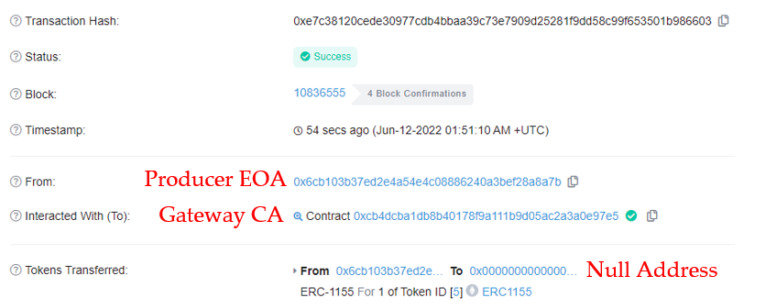
Transaction results when sending expired tokens.

**Figure 17 sensors-22-05153-f017:**
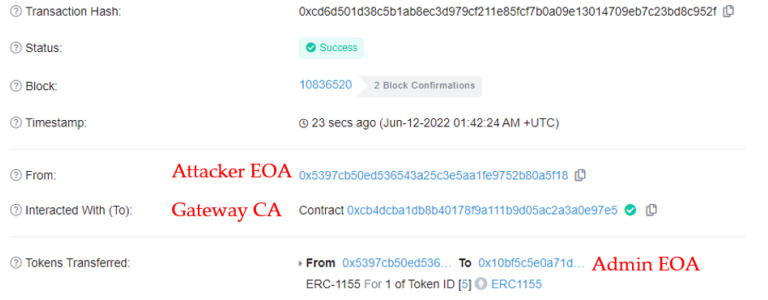
Transaction results when sending tokens exceeding the threshold.

**Figure 18 sensors-22-05153-f018:**
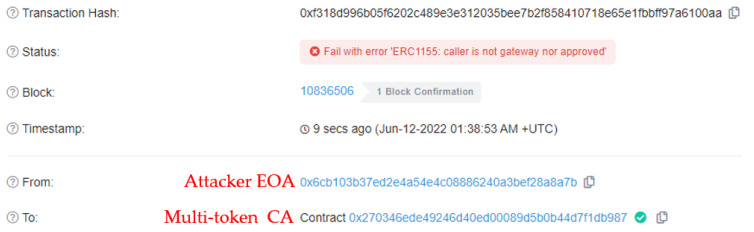
Transaction results when bypass the gateway contract.

**Table 1 sensors-22-05153-t001:** Compare related studies.

Feature	[10,11]	[12]	[13]
Tokens Tracking	O	O	O
Provide tokens to end consumers	X	O	O
Using ERC-1155	X	X	O
Preventing the distribution of invalid tokens in advance	N/A	X	X

**Table 2 sensors-22-05153-t002:** Standard for blockchain token.

Platform	Fungible Token	Non-Fungible Token	Multi-Token
Ethereum	ERC-20	ERC-721	ERC-1155
Klaytn	KIP-7	KIP-17	KIP-37

**Table 3 sensors-22-05153-t003:** Possible token standards.

Type	Fungible Token	Non-Fungible Token	Multi-Token
Quality inspection certificate	X	O	X
Pass certificate	O	X	X
Quality inspection certificate and Pass Certificate	X	X	O

**Table 4 sensors-22-05153-t004:** Ethereum Multi-token standard ERC-1155 interface function list.

Function	Parameter	Remark
balanceOf	owner, id	Returns the amount of tokens of token type id owned by owner
balanceOfBatch	owners, ids	balanceOf batch run
setApprovalForAll	operator, approved	Grant or revoke permission to transfer tokens to operator
isApprovedForAll	owner, operator	Verifies whether the owner has transferred authority to the operator and returns the result
safeTransferFrom	from, to, id, amount, data	Transfers amount tokens of token type id from
safeBatchTransferFrom	from, to, ids, amounts, data	safeTransferFrom to batch run

**Table 5 sensors-22-05153-t005:** List of EOA, CA used in the experiment.

Name	Account Type	Account Address
Admin	EOA	0x10BF5C5E0A71deb67c90f111209C435eAC3bAea9
Producer	EOA	0x6CB103B37eD2E4a54E4C08886240A3bEf28A8a7B
Consumer	EOA	0x70aEaD17350048Cada92514D55802D61b2126606
Attacker	EOA	0x5397Cb50eD536543A25C3E5Aa1fE9752B80A5F18
Gateway Contract	CA	0xcb4dCba1dB8B40178F9A111b9D05Ac2a3a0e97E5
Multi-token Contract	CA	0x270346ede49246d40ed00089D5b0B44d7F1Db987

**Table 6 sensors-22-05153-t006:** Transaction data used in the first scenario.

Parameter	Data Type	Data
tokenContract	Address	0x270346ede49246d40ed00089D5b0B44d7F1Db987
from	Address	0x6CB103B37eD2E4a54E4C08886240A3bEf28A8a7B
to	Address	0x70aEaD17350048Cada92514D55802D61b2126606
id	uint256	1
amount	uint256	1
data	bytes	Null

**Table 7 sensors-22-05153-t007:** Transaction data used in the second scenario.

Parameter	Data Type	Data
tokenContract	Address	0x270346ede49246d40ed00089D5b0B44d7F1Db987
from	Address	0x5397Cb50eD536543A25C3E5Aa1fE9752B80A5F18
to	Address	0x70aEaD17350048Cada92514D55802D61b2126606
id	uint256	1
amount	uint256	1
data	bytes	Null

**Table 8 sensors-22-05153-t008:** Transaction data used in the last scenario.

Parameter	Data Type	Data
from	Address	0x5397Cb50eD536543A25C3E5Aa1fE9752B80A5F18
to	Address	0x70aEaD17350048Cada92514D55802D61b2126606
id	uint256	1
amount	uint256	1
data	bytes	Null

**Table 9 sensors-22-05153-t009:** Analysis of gas by ERC-1155 Type.

Function	Gateway Contract	ERC-1155 A	ERC-1155 B	ERC-1155 C
deploy	1,156,753	2,209,376	3,043,979	2,489,208
safeTransferFrom	74,915		71,295	58,353
mint	N/A	77,594	83,048	53,549

## Data Availability

Not applicable.

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
