# Peer review of "Blockchain Token-Based Wild-Simulated Ginseng Quality Management Method"

_sensors, 2022, doi:10.3390/s22145153_

Round 1
Reviewer 1 Report
The manuscript represents an interesting topic by which the authors suggest using blockchain tokens for wild-simulated ginseng quality management. However, the quality of presentation and the scientific soundness of the manuscript are questionable.
1) The introduction is very short. It lacks research background. Authors are suggested to add more discussion about food supply chain traceability and the associated issues that this manuscript tend to solve.
2) The innovation and novelty of the presented study is not very clear, authors can add a list of contributions to the introduction section.
3) The study addresses the problem of quality management using blockchain by considering the wild-simulated ginseng as a case of study. A justification of why selecting the "wild-simulated ginseng" specifically among all other food is required to shed light on the importance of the study.
4) Section 2 is lengthy and it mixes some background information with the proposed model, which makes it hard to read. Authors can consider splitting this section.
5) There exist no related works section in the manuscript, and this made it difficult to identify the novelty of the presented study. It is important to for a research paper to include a number of relevant works and how does this work differ from them.
6) The technical aspect of the presented manuscript is weak; there is no formal presentation of the smart contracts , the concept of threshold logic, the system workflow,...etc.
7) How are tokens tracked? how to validate tokens?
8) The experiments seems to be unlinked with the proposed tokenization system. Moreover, the experimental results were not validated. For example a cost (gas) analysis of the smart contracts can be provided to show the feasibility of the system.
Author Response
Point-to-point responses to comments are written on the first page of the pdf. Thanks for the nice review comments.

Reviewer 2 Report
The authors have proposed a blockchain-based approach using tokens for food supply chain management. The article has reviewed the various blockchain tokens and their application for quality management and further presented the experimental work to justify their proposed method. There are some minor corrections that can improve the article:
1. The article should also cover the previous works conducted for the food supply chain.
2. The justification for using the proposed model is weak and some more elaborate justification is required. It should explain explicitly that how the existing models are not able to handle the food supply chain quality management and what are the loopholes in the existing studies.
Author Response

(The authors gave the same response as above.)

Reviewer 3 Report
Regarding the content, I do not have any changes to recommend, it makes a good literary review to support the relevance of the problem to be studied and a good structuring of the content, it uses the correct methodology for this type of study and it is a consistent and well-detailed methodology to give significance to the results they show, makes a good discussion of the results with respect to the studies carried out previously, and marks the conclusion obtained well.
Although I advise looking at these things:
Never two sections without a paragraph of text in between. You should put a couple of lines describing/naming the subsections you are going to deal with within that section. You must correct this between lines 49-50 and 50-51.
In the section “4. Discussion”, you must also add a comparison of the results achieved in your study with the results achieved in other studies. This will also help to have more academic references in your study, since there are few academic references in the “References” section.
In the section “5. Conclusions”, it is necessary to develop a deeper analysis of the implications of the study, before future work.
The references in the 'References' section must follow the model set by the journal. You must correct the errors that exist. Look at this in the template.
And “Author Contributions” section must follow the model set by the journal. Look at the template.
Author Response
Point-to-point responses to comments are written in Chapter 1 of the pdf. Thanks for the nice review comments.
